# Language Education for Forced Migrants: Governance and Approach

**Mike Chick [1],*** and **Iona Hannagan-Lewis [2]**

1   School of Humanities, University of South Wales, Pontypridd CF37 1DL, UK
2   Welsh Refugee Council, Cardiff CF24 1NJ, UK; ihlewis@live.co.uk
*   Correspondence: mike.chick@southwales.ac.uk; Tel.: +44-1443-480480

**Abstract:** This article stems from research conducted into the barriers to education, employment and language learning for refugees resettled into the convergence areas of Wales, UK. The authors consider that effective language programmes should play a key role in migration policies designed for multilingual, multicultural societies. The provision of English language classes for speakers of other languages (ESOL) ensures equality of opportunities, and in doing so, enriches the culture of our societies. By highlighting the challenges to language learning faced by refugees on the Syrian Vulnerable Persons' Resettlement Scheme (VPRS), this article draws attention to the fact that government directives for language provision commissioned under VPRS often do not sufficiently meet the needs of teachers and learners at grassroots level. Recommendations for greater flexibility in the organisation of ESOL provision for those resettled under VPRS are put forward. While this paper focuses on the specific case study of VPRS participants in Wales, it is hoped that recommendations around changes to policy and practice in language learning may be applicable to teachers, policy-makers, and community organisers working at the nexus of language and migration.

**Keywords:** language policy; integration; ESOL; forced migrants; Nation of Sanctuary; refugees

---

## 1. Introduction

In writing this paper, we would like to draw attention to the complexities involved in designing, organising and delivering a programme of language education for refugees on the Syrian Vulnerable Persons' Resettlement Scheme (VPRS). In light of the debates around the role of language in migration policies (e.g., see Khan and McNamara 2017), the paper draws on research into the particular experiences of those resettled in areas of Wales outside of the towns and cities into which asylum seekers are typically dispersed[1]. Specifically, the research has focused on the experiences of resettled refugees in the 'convergence areas' in Wales—local authorities of low GDP and which are in receipt of structural funds from the European Union. Martin-Jones notes that there is much to be learned from research into the field of "applied linguistics of globalisation", and from the study of language education programmes in particular (Martin-Jones 2015, p. 257). This paper aims to illuminate the experience of refugees resettled in these spaces with regard to the formal and informal ways in which their language education has been supported. The paper will detail the practical challenges of providing language education and draw attention to ways in which resettled refugees' experiences of language learning are shaped by the intersection of migration, integration and education policies and the availability of educational infrastructure in the host communities.

---

1   Under the 1999 Immigration and Asylum Act.

In the context of contemporary discussions on migration, the terms 'migrant', 'refugee', and 'asylum seeker' are often used interchangeably. This blurring of terms hides significant differences in the level of support and access to services afforded to people on the basis of migration status—whether refugee, asylum seeker, or a person who has migrated to the United Kingdom for the purposes of work, travel, or family. Even within the refugee population, the experiences of those who have been resettled in the United Kingdom as part of a managed resettlement scheme differ significantly from those who have gained refugee status following a claim for asylum. This article will therefore begin with a clarification of terms and a brief comparison of the support offered to those on VPRS, in contrast to those who have claimed asylum in the United Kingdom. It will then present original research into the provision of language learning to refugees on VPRS in the convergence areas in Wales and argue that current provision can be made more effective if greater flexibility is adopted in provision. An approach to language learning which places greater emphasis on learner agency will be proposed.

## 2. Context

### 2.1. Asylum and Refugee Resettlement in the United Kingdom

While the Syrian Vulnerable Persons' Resettlement Scheme has received much media and political attention, the scheme is only one way in which those who have fled persecution may have gained right to remain in the United Kingdom. The right to seek asylum is a human right enshrined in international law and is backed by the United Nation's 1951 Convention Relating to the Status of Refugees (the Refugee Convention) (UNHCR 1951). Under the terms of the Refugee Convention, a refugee is defined as any person who:

> *Owing to well-founded fear of being persecuted for reasons of race, religion, nationality, membership of a particular social group or political opinion, is outside the country of his nationality and is unable or, owing to such fear, is unwilling to avail himself of the protection of that country; or who, not having a nationality and being outside the country of his former habitual residence as a result of such events, is unable or, owing to such fear, is unwilling to return to it (Article 1A (2): ibid.).*

Under UK immigration law, 'refugee status' refers to a 5-year, limited leave to remain in the United Kingdom that is awarded to a person if they are deemed to have a legitimate claim to asylum according to the terms of the aforementioned Convention. While a person is in the process of having their claim for asylum considered, they are typically referred to as an 'asylum seeker'. As an asylum seeker, a person has no recourse to public funds, is generally barred from seeking employment (unless their profession is on the shortage occupation list) and is liable to be detained at any point (Right to Remain 2018). If the person seeking asylum is destitute and has nowhere to live, then they will be entitled to asylum support of £36.95 per week and offered accommodation on a no-choice basis in a dispersal city within the UK. In Wales, the dispersal cities are Cardiff, Newport, Swansea and Wrexham (Welsh Government 2019a). If a person is then granted leave to remain following a successful claim for asylum, they may be termed a 'refugee' in common parlance, although it is notable that 'refugee status' is only one form of leave to remain which may be granted to a person following a successful claim for asylum (Right to Remain 2018).

Compared with those who attain refugee status following an asylum claim, significantly fewer people attain leave to remain in the UK following participation on a managed refugee resettlement scheme, with resettled refugees making up just over a quarter (27%) of the total number of those granted asylum in the UK between 2014 and 2018 (Sturge 2019). Unlike the UK's standard asylum procedure outlined above, under a resettlement programme refugees are typically given refugee status (or another form of leave to remain) while abroad, prior to being brought to live in the UK (ibid.). VPRS is the largest-scale resettlement scheme currently underway in the United Kingdom—in June 2019 the Home Office announced that it had resettled approximately 16,000 refugees since the scheme's inception in 2014 (Home Office 2019). VPRS is due to terminate in 2020, however the Home Office recently announced its intentions to continue resettling refugees beyond this date, with the launch

of one "global" resettlement programme which will consolidate the various managed resettlement schemes (including VPRS, the Gateway Protection Scheme, and the Vulnerable Children's' Resettlement Scheme) (ibid).

Under VPRS, refugees are offered considerably more support than those who attain refugee status through the asylum route. The scheme offers tapering funding to local authorities to meet the costs incurred by resettling refugees—from full funding in the first 12 months of a refugees' resettlement, to £1000 per individual in year 5 (Home Office 2018). In addition to this funding is the £10m pledged by the UK Government to "enhance the English language skills of adults to improve their resettlement and integration experience and employability" (Home Office 2017, p. 6).

By contrast, those who gain refugee status following an asylum claim do not have access to comparable levels of funding or support on a local authority level. The differences between the level of support offered to those on VPRS to that offered to those on the asylum route has led to the observation that the UK has, in effect, "a two-tier" system of support for refugees (APPG on Refugees 2017). Reflecting on this difference, an interviewee in our study (a caseworker for VPRS participants) remarked that:

> "The support for the refugees on the managed programme is gold standard. We help with so many aspects of resettlement such as getting homes ready, placing children in schools, registering with the local health centre, helping them with the benefits system ... people granted refugee status through the asylum process have a very different experience, indeed". (J.S.)

All 22 local authorities in Wales are now participating in the VPRS and by January 2019 close to 1000 refugees had been resettled across the country (Welsh Government 2019b). Of key importance to this article, is that VPRS has resettled refugees in locations outside of the dispersal cities listed above. This means, in some cases, refugees are being resettled in areas with little or no prior experience of refugee resettlement, and thus in areas that have negligible established infrastructure (such as language classes) to support migrant integration. In focussing on such areas, this article documents some of the challenges emerging at ground-level as local authorities adapt to the needs of new refugee populations.

## 2.2. Refugee Integration in Wales

The authors acknowledge the complexity of defining the concept of integration. The experience of feeling 'well-integrated' into one's community is highly subjective, and the factors which contribute to such a feeling will likely vary substantially between people and across context and time. Moreover, as Spencer and Charsley note, the term carries with it an exclusionary potential, indexing an assimilationist, one-way conceptualisation that over-emphasises the newcomer's responsibility for integrating, rather than that of the community (Spencer and Charsley 2016, pp. 3–4). Despite these complexities, 'integration' is the term used by many of the many policy documents, guidelines, and publications relating to the VPRS participants with whom we were researching. As such, for the purposes of our research it was necessary to first establish a working definition of the term. In order to do this, we have drawn primarily on the aforementioned work of Spencer and Charsley (2016) and Ager and Strang (2008), both of whom have developed models of integration which conceptualise integration as manifold, rather than one-way. For Spencer and Charsley, integration is a two-way, evolving process between the newcomer and community, which develops across several domains; social, structural, cultural, civic and political, and that of identity (Spencer and Charsley 2016, p. 4). Similarly, Ager and Strang's conceptual framework on refugee integration has ten 'domains' across four tiers (see Figure 1, below).

**Figure 1.** Ager and Strang (2008, p. 170).

Employment, housing, education, and health appear in the framework as both markers of and means towards achieving integration. The aspect which link the foundational and means/marker tiers is that of social connection. Refugee access to social connection is influenced by the 'facilitator' tier, in which knowledge of the host-country language and culture, as well as a sense of safety and security, can either facilitate or inhibit a refugee's access to social connection, and thus to the four means/markers of integration (Ager and Strang 2008, p. 170).

Heckmann notes that understanding 'barriers' to integration should form "an integral part of integration research" (Heckmann 2006, p. 14). Furthermore, and of particular relevance to this article's focus on the importance of language learning, is that Ager and Strang's analysis situates the 'facilitator' tier (including language and cultural knowledge) as a key site in which "actions could serve to facilitate (or constrain) local integration"—that is, in which barriers to integration can occur if the state, community, or responsible parties do not act to remove them (ibid., p. 181–82).

While immigration and nationality are matters reserved for Westminster administration[2], many of the sectors contributing to refugee well-being and integration are devolved to Welsh administration. These include health, housing, education, and travel. The devolved government of Wales has, in recent years, developed several policy initiatives to support refugee integration and subsequently there are significant policy contrasts with England in terms of approaches and openness to immigration. For example, the *Prosperity for all National Strategy* (Welsh Government 2018c) and the *Well-Being of Future Generations* Act (Welsh Government 2015) promote the vision of equality for all in Wales. Moreover, Wales has recently outlined its vision of Wales as a 'Nation of Sanctuary' for all in its 'Nation of Sanctuary—Refugee and Asylum Seeker Plan' (Welsh Government 2018b). The difference in approach to immigration governance between England and the devolved administrations was noted in a recent report by the Office of the High Commissioner for Human Rights which observed that there were more "human rights-based approaches to immigration among government authorities in the devolved nations" (Office of the High Commission for Human Rights 2019, p. 16).

Regarding language education, in 2014 the Welsh Government distanced itself from the Westminster administration through the introduction of an English for speakers of other languages (ESOL) Policy for Wales (updated in 2018) and its commitment to "continue to fund ESOL as an Essential Skill" (Welsh Government 2018a, p. 15). The most recent estimate suggests that there are, approximately, eight thousand learners enrolled on ESOL courses across Wales, a figure that does not include informal provision delivered by charity, faith-based and community groups (Welsh Government 2014). In Wales,

---

2     Under the Wales Act 2017.

ESOL is funded as an "essential skill" up to Level 2, which means it is one of the skills, along with essential employability skills and digital literacy, that providers, such as colleges of further education, are required to address first. The Adult Learning policy also includes a pledge that ESOL classes will be free to all learners "up to the level of functionality" (Welsh Government 2017).

In sum, the political context in Wales can be considered to be broadly sympathetic to achieving positive outcomes for refugee integration, particularly in contrast to the range of government measures aimed at reducing migration to the UK that collectively came to be known as the 'hostile environment' policy (House of Lords Library Briefing 2018). The Welsh Government's commitments to supporting the integration of refugees and asylum seekers, as evidenced by the aforementioned Refugee and Asylum Seeker delivery plan and the ESOL policy for Wales, provide a strong context for improving integration outcomes for migrant communities in Wales. Nevertheless, as this article will show, there remain challenges to providing effective language education programmes to forced migrants, especially to those on managed resettlement schemes who have been resettled far from the dispersal cities.

## 3. Methodology

The data discussed in this article is drawn from a study which took place between October 2017 and February 2019. The study was funded by the European Social Fund under the Knowledge Economy Skills Scholarships (KESS II) and conducted by the University of South Wales in partnership with the Welsh Refugee Council. The study adopted a mixed methods case study approach, with five local authorities consenting to participate as case study areas. The participating local authorities were all drawn from the convergence area, and all had little prior experience of refugee resettlement.

Refugees resettled on the VPRS have been selected for resettlement by UNHCR and the Home Office based on criteria assessing their vulnerability. This means that the participants might well be people identified as "at risk" such as being survivors of violence, individuals with disabilities or those with physical protection needs (Bolt 2018, p. 46). As such, participants on the VPRS are afforded a degree of anonymity and protection by the local authorities charged with their care, and access to the scheme's participants is understandably restricted for legal and safeguarding reasons. In order to gain participants' consent to contribute to the study, it was therefore necessary to first gain the agreement of those acting as gatekeepers—VPRS co-ordinators, caseworkers and translators in participating local authorities. Participant recruitment was then conducted via these networks, with all resettled refugees over the age of compulsory education (16) invited to participate in a paper survey, which was translated into Arabic. As such, the sampling method adopted for the refugee sample can be broadly termed a 'convenience sampling' approach (that is, the sample was chosen on the basis of its availability), which was adopted owing to the limited access the researchers had to the resettled refugee population (Bryman 2016, p. 187).

The survey gathered data relating to participants' first language literacy, education and employment history, current employment and study, and aspiration. Forty-five valid questionnaires were returned. The data obtained formed the basis of focus groups, which were conducted in each case study area with the aid of translators. A total of 58 VPRS participants over the age of 16 participated in the focus groups, which were audio recorded and transcribed. In addition, interviews were conducted with 26 stakeholders across the case study areas. A theoretical sampling approach was adopted for the interviews, with data collected on an iterative basis (ibid., p. 410). Stakeholders targeted for interviewing were those with responsibility for the organisation or delivery of education and employment services to participants of the Syrian Vulnerable Persons' Resettlement Scheme in each of the case study areas.

Qualitative data analysis was conducted on the transcriptions of the focus groups and interviews. The transcriptions were coded by hand and by utilising qualitative data analysis computer software, and the codes compared to develop theories grounded in data analysis. This method of qualitative data analysis can broadly be termed 'thematic analysis', (ibid., p. 584). We sought to remain as faithful as possible to our perception of the 'spirit' in which participants contributed to the study. This meant

reviewing texts with sensitivity to context, non-verbal utterances and tone of voice, as well as the words themselves. However, all research involves a degree of researcher bias, and therefore the themes that we have focussed on inevitably reflect the researchers' own positionalities and subjectivities.

All participants, whether refugee, service provider, or otherwise, have consented to participate in the study on the understanding that their data will be anonymised, and all initials used in this article are pseudonyms. For direction on ethical considerations and procedures, the guidelines offered by BERA (2011) provided a useful framework as to what issues needed to be taken into account. The protocol was also given approval by the University Ethics Committee under the identification code C80815. Moreover, all references to place names, including the names of regions, towns, groups, or institutions, have been avoided. Finally, of note is that focus groups and interviews took place in a variety of languages [English, Arabic, and Welsh], but all excerpts are presented here in English translation. The use of interpretation and translation, where the role of language is crucial, has major implications for methodological rigour and validity in qualitative research. For example, the researcher has no control when the participants and interpreter are communicating in a language the researcher does not speak. Validity may be threatened when questions or responses are elaborated, limited or even erroneously translated. What is more, interpreters may well make decisions on the content of participants' utterances. Selective translation, therefore, raises issues of collaboration, power, and representation, all of which need to be considered in taking a reflexive stance towards qualitative data analysis. Research by Edwards (1998), Kapborg and Berterö (2002) and Temple and Edwards (2011) offer useful guidelines for recognising and acknowledging such issues.

Finally, it should be noted that owing to the use of a convenience sampling approach to data collection, the findings of this study cannot be considered representative of the entire population of refugees resettled in Wales. The findings detailed in this article should thus be considered as a snapshot of the experiences of individuals resettled across the case study areas. Nevertheless, by highlighting the experiences of these individuals, the authors hope to provide insight into some of the barriers and challenges faced by refugees in accessing language learning post-resettlement. All suggestions for improvements to existing provision are drawn from the views and opinions of our research participants.

## 4. Findings

The following section briefly summarises key aspects of the UK Government Home Office advice for ESOL provision, provided to local authorities taking part in the managed resettlement programme (Home Office et al. 2016). The guidance will then be contrasted with findings from the data collected for this study. The discussion that follows suggests that greater clarity about how ESOL should be organised and provided would be beneficial to local authorities that do not have a developed ESOL infrastructure. Moreover, it is argued that alternative models of ESOL provision need to be considered for VPRS participants, in addition to the model of accredited college ESOL provision.

Throughout the Home Office ESOL guidance (ibid.) for the VPRS, it is emphasised that English language is "an essential element of their (refugees') integration and journey towards self-sufficiency" (ibid., p. 4). Indeed, in referring to additional funding given over to ESOL, language support is framed as a way to "improve [resettled refugees'] resettlement and integration experience and employability" (ibid., p. 7). Thus, developing English language competency is considered to be key to improving integration outcomes, securing employment, and becoming independent from state support. As a consequence of this emphasis on language development, local authorities taking part in VPRS receive funding from the Home Office and are required to "make appropriate English language support arrangements for resettled people" (ibid., p. 4).

As our data will show, providing appropriate language support to individuals with vastly different learner backgrounds, different levels of competency and literacy, often rehomed in geographically dispersed areas, is a complicated and labour-intensive task. The Home Office guidance notes allude to this complexity, stating that the aim of ESOL funding is to "enable learners to access accredited ESOL provision, which means formal classes leading to an ESOL qualification" as well as "pre-entry

level ESOL provision that does not lead to an accredited qualification where appropriate" (ibid., p. 7). Moreover, the local authority must:

> "effectively identify an individual's ESOL requirements and be responsive to those needs through the most appropriate delivery arrangements and range of providers within a local area". (ibid., p. 7)

With regards to the number of hours of ESOL offered to VPRS participants, the Home Office (2018) funding instruction document for local authorities recommends that, where appropriate, a minimum of 8 hours a week of language education provision should be available (Home Office 2018, p. 19).

The guidance also acknowledges that there are numerous issues that can affect an individual's ESOL requirements. Indeed, it is stated that funding can be allocated to overcome barriers to access such as childcare, transport, childcare support, disability, unfamiliarity with formal learning, transport to classes, and the availability of existing local ESOL infrastructure (ibid., p. 25). Despite this, our research findings show that these barriers continue to prevent VPRS participants from accessing ESOL across the case study areas. The following section records issues around access to language learning experienced by refugees resettled across the case study areas.

(i)　　Number of hours of ESOL

The refugee survey sample (n = 45) was asked to state the number of hours of ESOL they accessed per week. This data is presented in the scatterplot (Figure 2, overleaf). In order to protect anonymity, case study areas are numbered (from 1 to 5), rather than named. It should be noted that the maximum number of hours of formal ESOL available to VPRS participants in any case study area was 16. In order to account for the outliers at the higher end of the scatterplot, the authors consider that survey respondents may have included some hours of self-study in their responses. Nevertheless, data on the hours of formal ESOL available to VPRS participants across the case study areas affirmed the general picture presented by the scatterplot—that there is a marked inconsistency in the number of hours available to VPRS participants resettled across Wales. In case study areas 2, 3, and 4, there was considerable discrepancy in the number of hours of ESOL refugees were able to access, even among those who were resettled in the same local authorities.

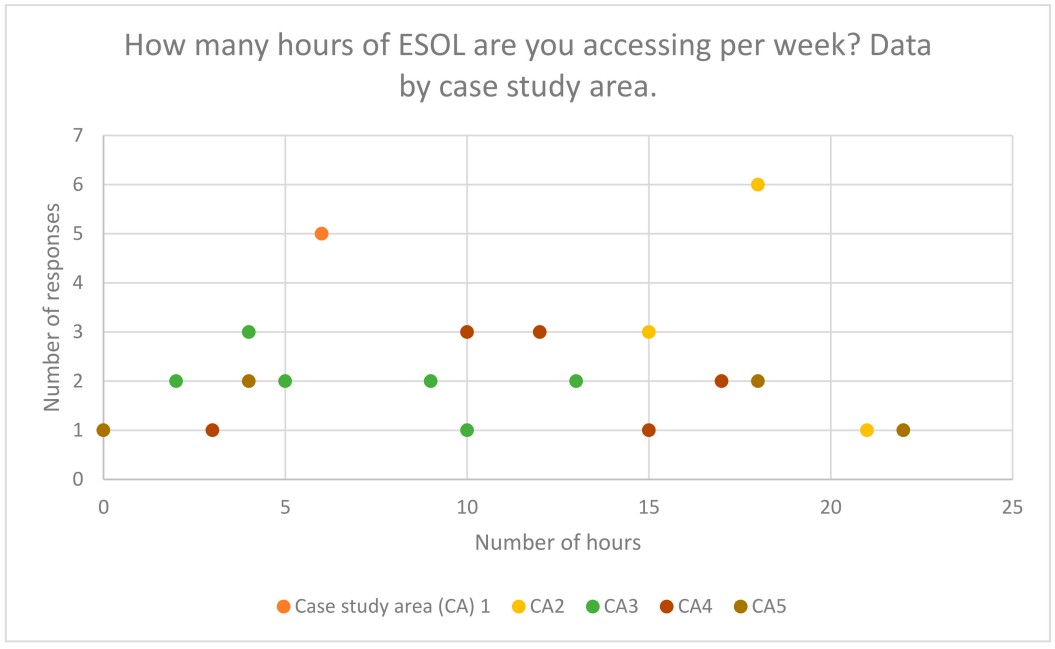

**Figure 2.** Scatterplot indicating responses to the question 'How many hours of ESOL [English for speakers of other languages] instruction are you accessing per week?' Data shown by case study area.

In response to the question, 'Do you feel you are getting enough hours of ESOL per week?' 24 survey respondents reported that they felt they weren't getting enough hours of instruction, while 17 reported that they were. The main reasons for the paucity of hours of ESOL provision in some areas are explored below.

(a)   Access to childcare

Lack of childcare provision as an obstacle to participation in language learning was most often cited as a barrier in case study area 5. All the women resettled to this area had had their participation in language learning affected by the fact that the local authority had not been able to secure funds to subsidise childcare.

1    IHL: What's the problem?
2    OL: For babies I cannot go college but for baby.
3    IHL: Okay
4    Interpreter: It's the problem with the for the nursery for the children there is no fund for the nursery so they can't leave their children. [ . . . ]
5    Int: In [area 2, the council] are paying for the nursery
6    IHL: So [area 5] they can't have no money for childcare and [area 2] they do?
7    Int: Yes. [ . . . ]
8    IHL: Do all the women have this problem?
9    Many voices: Yes, yes
    (Excerpt from focus group, case study area 5)

Female participants from case study area 5 highlighted the impact of a lack of childcare, as below:

1    Interpreter: She said that is the biggest thing that they are stuck in and they want to go on the bus and the bus driver ask them something and they want—they don't know how to reply
2    So yesterday somebody was talking to her whatever she was very [shrug shoulders, shake head]
3    but her daughter was helping her, her younger daughter to translate [ . . . ]
4    She like to talk with her neighbour as well but the language barrier [ . . . ]
5    I haven't got opportunity to learn, I haven't got opportunity to learn that's what she's saying, haven't got opportunity to learn the language
    (Excerpt from focus group, case study area 5)

As the above excerpt shows, not only did a lack of childcare impact women's participation in language learning, but their ensuing low proficiency in English impacted their independence and mobility. The participant in question reported feeling 'stuck in', wanting to travel and use the bus, but unable to communicate with the driver. This experience of isolation is further captured in this excerpt from a survey (below), in which a participant who cannot attend language classes due to a lack of childcare expresses her frustration at being unable to acquire sufficient language to help her children with their homework, or to communicate with people in her community (Figure 3, overleaf):

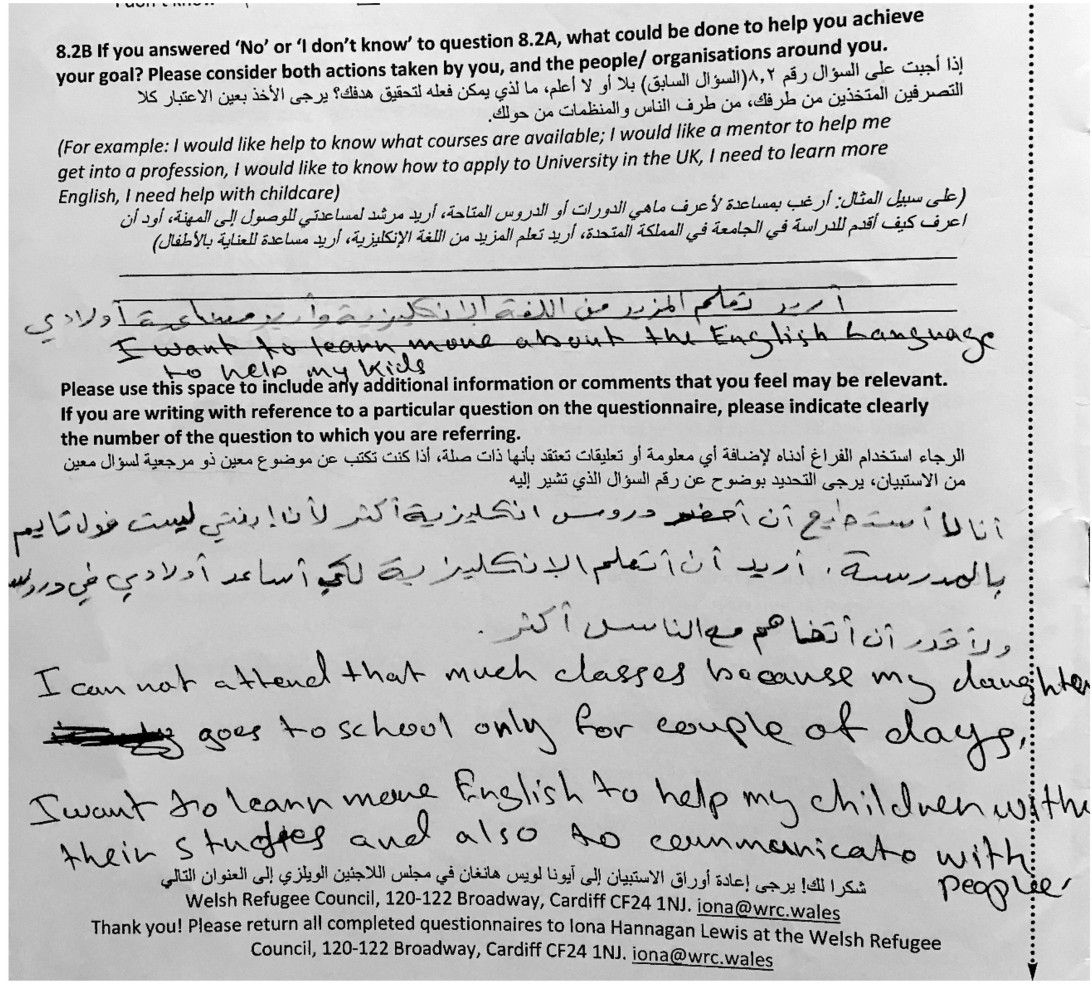

**Figure 3.** Photograph of page from a questionnaire. The Arabic translates as: "I want to learn more about the English language to help my kids/I cannot attend that much classes because my daughter goes to school only for a couple of days. I want to learn more English to help my children with their studies and also to communicate with people."

Case study area 5 was exceptional among the areas under study for not having secured adequate funds to support childcare for participants. However, among the case study areas that had secured funds towards this item, there was notable uncertainty about the sustainability of childcare provision once the VPRS funding had ceased. For example, in case study area 2 (the "larger county" with which respondents in case study area 5 were comparing themselves), the participants and reported a lack of certainty as to the future of childcare provision after the first year of resettlement (in which, as detailed in Section 2.1, full funding is provided):

1    IHL:          Is there childcare? Is there childcare?
2    KA:          Here in college?
3    IHL     Yeah
4    [many voices]: No [ … ]
5    Interpreter:    No there is not, not in college no, no it's not it's only put them in private nurseries
6    IHL:          So, does- does private nurseries get paid by the council?
7    Int:      Yep- yes
8    IHL      So that means you can go to college?
9    Int:      Yes
10   IHL:         Good. Other areas=

| 11 | Int: | = sorry |
|---|---|---|
| 12 | | It's only for one year, yes? |
| 13 | FA: | Yes |
| 14 | Int: | Only for one year [from council] |
| 15 | IHL | Ah, and then after you= |
| 16 | Int: | =till now we don't know. |

(Excerpt from focus group, case study area 2)

A similar uncertainty about the organisation of childcare beyond the first year of resettlement was found in case study area 1, as in the below excerpt with the local authority community learning manager ('VT'). As this excerpt shows, even in those case study areas which had secured funding for childcare, there was little clarity on how childcare would be offered to support VPRS participants' studies following the termination of programme funding.

1    IHL: So, beyond VPRS, will they have access to childcare as well?
2    VT: I don't know to be honest, I'm assuming that it is continuing
3        because they're still attending the ones that needed childcare
4        even though they're beyond the 12 months
5        so I'm assuming that the funding is still there.
(Excerpt from interview with 'VT' community learning commissioner, case study area 1)

(b)   Local infrastructure and travel

Local availability of ESOL also played a role in determining how many hours of language learning resettled refugees were able to access, and the cost to them of accessing classes. In areas which did not have a large range of ESOL provision in existence prior to VPRS, we found that refugees would frequently have to travel, sometimes to another local authority area, in order to access classes. This was the case in area 5, where the male learners were travelling to study at the college in area 2. In the following excerpt one participant from case study area 5 adds up the cost of travel for himself and his family:

1    AB: I give you example:
2    when coming to college we have to uh buy a weekly ticket
3    £14.50 and for one person
4    if we come with his wife uh £29 a week-weekly
5    and if they want to come as a family to [xxx] they have to pay as a family ticket £12 this will cost I'm sure more than £150 monthly, yeah?
(Excerpt from focus group, case study area 5)

While some of those in resettled in area 3 were able to access up to 15 h of ESOL per week at the local college, those in in another district of the same area could only access up to 4.5 h at the local community centre—despite the fact that provision at both venues was run by the same further education provider. When the issue was raised in the focus group, it transpired that, while caseworkers had requested for the refugees resettled far from the college to be provided with bus passes, this had yet to happen:

1    Caseworker: The council have been in talks with [the bus company] for over well since last September
2    and the reason is [the council] only wants to fund quarterly season tickets uh
3    but unfortunately the bus company doesn't do season tickets.
4    Also [the council] only wants to allow travel in the area not the further area as well so it's hard to get that organised.
(Excerpt from focus group in case study area 3)

An interview with the college ESOL co-ordinator in area 3 highlighted the difficulty of meeting the needs of refugees resettled far from the college, as the college are already working 'full time':

1    EC: [ … ] the ESOL provision is here. They're talking about us giving them extra lessons but we are working full time.
2    When else do they want us to teach them?
3    Do they want us to go to their houses? They being the council.
4    Because our provision is here.
5    Traditionally, we've got classes on a Tuesday in [this part of the local authority] because we haven't had as many students in [the other district] as we have [here].
(Excerpt from interview with ESOL co-ordinator, further education provider, case study area 3)

As indicated in the above excerpt, in areas in which there are relatively low numbers of English language learners there were deemed to be insufficient numbers of ESOL learners to justify the commissioning of bespoke full-time language courses. The lack of learner numbers was also problematic in case study 4 as in this excerpt from an interview with the co-ordinator of a voluntary English language teaching initiative (VT) and the VPRS co-ordinator, working in the local authority (RA):

1    VT: I think we do need 3 different levels, I think we have people of 3 different levels.
2    We have one class that are learning the alphabet from scratch,
3    but we've also got [ ] who is struggling to make the figures and do the alphabet. [ … ]
4    I really think we do need the 3 classes.
5    RA: The only issue is that [national education provider] say they need 8 for a class [ … ]
6        There are obviously people out there who need it.
(Excerpt from interview with voluntary organiser ('VT') and VPRS co-ordinator ('RA') case study area 4)

Thus, refugees resettled in areas far from existing ESOL provision are caught in a difficult situation—while they are in need of ESOL, education providers consider that it is not cost-effective to commission full-time ESOL in areas with low learner numbers. However, in addition to inhibiting the number of hours of ESOL available to refugees in some areas, the issue of cost-efficiency was also a factor in determining the range of class levels available.

(ii)   Levels

A recent report for the UK Government Department for Education found that teaching "someone who may have no literacy alongside those that have substantial experience of prior education is extremely challenging" (Higton et al. 2019). Likewise, in this study, much frustration concerned the lack of differentiated levels. As mentioned above, existing ESOL infrastructure varied in size and scope in the case study areas in this investigation. In those areas with a less-developed suite of English language provision, learners were frequently grouped together in the same class. This caused marked dissatisfaction and even classroom tension among learners, as one VPRS participant stated:

1    And sometimes because there are people of a lower level [ … ]
2    they can't catch up quickly.
3    They need their own very low level.
4    Beginners level, from the letters [ … ]
5    they are struggling, they get angry, the class is tense, believe me, and a few problems happened in the class (VPRS participant).

For teachers, attempting to prepare some students for upcoming assessments while teaching literacy skills to others, class management becomes an unenviable task—as indicated by in the following interview excerpts:

| 1 | It's heart-breaking, impossible really, to turn anyone away from a class. |
| 2 | But the truth is that having pre-entry level students |
| 3 | who can't yet identify the letters of the alphabet |
| 4 | in the same classroom as learners at entry level 1 or 2 is a big problem for everybody (ESOL teacher) |

| 1 | You get people who don't even know the alphabet |
| 2 | and you get people who have been coming for two years and everyone becomes frustrated |
| 3 | and the lecturer is tearing her hair out. (ESOL manager) |

The financial conditions placed on colleges to open an ESOL class were frequently cited as the reason comprehensive, level-differentiated programmes were not being provided to some resettled refugees. One stakeholder explained the dilemma:

| 1 | They are all in the same class |
| 2 | and we've got a range of levels because some of them have been here for an additional year and more families are arriving |
| 3 | so there is a mix in the class. |
| 4 | It's the numbers, it's just not financially viable for the college to put them in separate levels. (ESOL manager) |

Thus, despite being on the same resettlement scheme, refugees resettled on VPRS across Wales are accessing an inconsistent patchwork of language education. The following section considers the question of who is responsible for the delivery of ESOL to resettled refugees in Wales.

(iii)    Organisation and Governance of ESOL for resettled refugees

As detailed in Section 2(ii), ESOL is funded by the Welsh Government as an Essential Skill and delivered via further and community education providers. In three out of five case study areas, resettled refugees generally had access to Welsh Government-funded ESOL at the local further education colleges (although, as detailed above, some were unable to access classes owing to the barriers of childcare, travel, and the availability of local classes). Those refugees able to access these classes had their linguistic competency assessed prior to being subsumed into the appropriate level within the college's existing classes. Those that were able to access sufficient hours of ESOL at an appropriate level tended to report satisfaction with the provision on offer. Furthermore, in these cases, there was a clear chain of accountability linking the students and colleges to outcomes outlined in the ESOL Policy for Wales:

> *"To [ . . . ] have systems that accurately capture the progress made by ESOL learners, reflecting the individual modes of learning, and the length of time necessary for ESOL learners to progress"*. (Welsh Government 2018a, p. 10)

However, in areas where there is no or little Welsh Government-funded ESOL infrastructure, our research revealed confusion on the part of the local authorities as to who held responsibility for devising and initiating a comprehensive programme of ESOL according to expectations outlined in the Home Office VPRS funding instructions. Interviews with local authority officials in two of the case study areas revealed that they felt unprepared and unsupported with regard to overseeing the language element of the scheme. One of the resettlement officers, concerned about their lack of experience in the field of language education, reflected that:

| 1 | It would have been helpful to have someone talk in lay-persons terms about what ESOL is, how it's provided, who provides it, what the accreditation is [ . . . ] |

The official was anxious about the linguistic component of the scheme, reflecting that:

| 2 | [ . . . ] when we first took families there were no clearly defined expectations about language- |

3     How much was enough? How much wasn't enough?

Perhaps reflecting the confusion as to who holds responsibility for ESOL for the resettled refugees in a devolved context, the same official directed their dissatisfaction towards the Welsh Government:

4     [ . . . ] at the start it would have been good to have spoken about the expectations and how ESOL is delivered,
5     and that could have come from Welsh Government.
      (Excerpt from interview with local authority resettlement officer)

Further education colleges, where the majority of ESOL in Wales is delivered, were also frustrated by the attempts to organise language classes for the newly arrived Syrians in the community. As explained above, colleges are tied by funding limitations and have had no contact with the Home Office about language provision for refugees on the VPRS. In one of the case study areas, the ESOL manager was clear that it was not their responsibility to organise ESOL for the resettled refugees, stating:

1     We are not responsible for organising the Syrian's ESOL
      (Excerpt from interview with ESOL manager at a further education college)

In another, the ESOL manager acknowledged the frustration of having what are effectively two parallel programmes of funded ESOL running concurrently, without clear guidance on which organisation is responsible for delivering the outcomes expected under VPRS funding guidelines:

1     It's really frustrating because we are supposed to be a college providing ESOL
2     and we are saying 'oh sorry we can't cater for you'.
3     Who else is supposed to be doing it?
4     If the college can't provide it.
      (Excerpt from interview with ESOL manager at a further education college)

Thus, where there were shortfalls in delivery (insufficient hours or access to appropriate levels), without clear understanding of the organisations or bodies responsible for delivering ESOL to resettled refugees, the researchers found little clear indication as to which organisation was responsible for ensuring that these were overcome. However, the authors noted several voluntary, informal community language initiatives and conversation classes taking place across the case study areas. In some cases, these community initiatives were playing a vital role in 'plugging gaps' in formal delivery, as the following section documents.

(iv)   Formal vs. informal language learning

In one local authority (case study area 4) with limited formal, accredited provision, a grassroots partnership initiative has developed between the local authority, a local church, and a university language department. The collaboration, now in its fourth year, has proven to be tremendously popular with the VPRS participants, providing the space and opportunity for them to improve their language competency outside of the confines of a conventional classroom. The collaboration has been recorded and detailed by Chick (2019).

In case study area 5, where women were unable to access classes due to childcare commitments, volunteer teachers were visiting the women at home to provide informal language tuition. For some learners such 'informal' classes are a social space where they can improve their language skills without facing the pressure of assessments, tests and so on. Indeed, for some learners—particularly with limited experience of formal educational environments, informal study may be a more appropriate learning environment. One interviewee recounted how 'Leila', a fifty-year-old, mother-of-six felt devastated for days after failing her end-of-term ESOL exams:

1     Leila is a wonderful woman,

2      hugely motivated to learn English—

3      not least so that she might be able to speak with her children and their teachers in English.

4      But she's never been to school.

5      She finds exams too challenging. She's awfully upset.
       (Excerpt from interview with community volunteer)

For some participants such as the learner described above, it was felt that an informal approach was more appropriate. Indeed, as we see from the data presented here, voluntary initiatives can provide a lifeline in the absence of formal provision. The considerable efforts of community activists in creating a culture of welcome and of providing vital integrative services should be recognised and applauded and can play a key role in ensuring the delivery of the Welsh Government's vision of a 'Nation of Sanctuary' for all. Nevertheless, it is the view of the authors that community organising should not be taken for granted, nor is voluntary provision alone sufficient to fill gaps caused by the absence of government-funded education programmes.

In a similar vein, some participants were eager to get out of a formal classroom environment and to start work. Some were disillusioned by a sense of spending hours in class, preparing for exams that seemed unrelated to their career aims or progress toward self-sufficiency:

1      I am not happy in school, because it is like an academic course,

2      so we are not young to study academic,

3      only we can improve our language in practical situations
       (Except from interview with VPRS participant)

This was reflected in focus group conversations, where some participants viewed working as a better way than ESOL classes to gain English language skills:

1      Interpreter: They say if they work help them to learn English [ . . . ]

2      Uh she say she know someone he don't study but he go to have work

3      and he never studied but his English now is very improved

4      IHL: Who is that?

5      Int: He is his cous-her cousin he live in London now and he got a job
       (Excerpt from focus group, case study area 1)

According to one caseworker, the ideal would be a scheme whereby people were able to learn English and work at the same time:

1      All of the refugees would want a job opportunity where they could work and learn English at the
       same time.
       (Excerpt from interview with VPRS caseworker)

However, at the time of writing, the only further education college in Wales offering integrated courses in ESOL with vocational qualifications was based in Cardiff, far from the case study areas into which our refugee research participants had been resettled. There were no known instances of refugees accessing work that included on-the-job language development and training.

It may be concluded from the above that, despite the Home Offices' seemingly comprehensive expectations for the delivery of language learning on VPRS, there have been significant complexities involved in delivering these expectations at community level. In some instances, local authorities have been insufficiently resourced—be it through funding, guidance, or expertise—to adequately address the complex needs of the VPRS participants that took part in this study. This is clearly true in the case study areas that do not have a wide range of ESOL provision already in place. Consideration of the data presented above has allowed us to reflect more deeply on the role of ESOL in managed resettlement programmes, and how it may be more effectively organised.

## 5. Discussion

The following section discusses the implications of this 'snapshot' of the reality on the ground for refugees on the VPRS in Wales. It is argued that a more flexible approach both to how ESOL is viewed and understood, as well as to how it is provided, is now necessary for refugees being resettled on VPR schemes. That the current model of ESOL provision can be improved is clear. We have identified several reasons for this, and these are detailed and discussed below.

### 5.1. Time

The question of how many hours' class a learner should receive is problematic for a couple of reasons. Firstly, there is very little research that provides convincing evidence on the optimum number of classroom hours for adult second language development, not least due to the numerous variables that affect the rate of acquisition. Secondly, what estimates do exist (e.g., see Welsh Government 2014), suggest that an adult learner starting from a pre-entry level of English (Pre-entry level is equivalent to A1 on the Common European Framework of Reference) and receiving ten hours of level-appropriate class each week would need nearly six years of study to complete ESOL level 1 (equivalent to level B2 on the CEFR). This is the level of functionality deemed desirable for entering the workplace (Welsh Government 2014, p. 4). The data we collected for this research found that the areas with little ESOL infrastructure were not able to provide ten hours of level-appropriate classes. Even based on such loose estimates, this would mean that funding of ESOL classes, as part of the VPRS, would likely cease before the participants reached level 1. Perhaps more pertinently, expecting adult learners, unused to formal educational settings, to attend five or more years of classroom-based instruction, in their attempt to achieve competency is unrealistic. New models need to be tried in which learners are given greater agency to pursue integration routes that are more closely related to their needs, abilities and expectations. This may mean, for example, facilitating work-experience programmes with elements of ESOL taking place alongside such work-entry schemes.

### 5.2. A Flexible Approach

Refugees on the VPRS are expected to receive the majority of their language education through attending formal classes " ... *so that the learners have the opportunity to gain an accredited qualification*." (Home Office et al. 2016, p. 4). However, as we have noted, restricting study to formal classroom settings alone is unlikely to provide the most effective language learning environment for all—and especially for those who have had little prior experience in a formal education environment. As Sidaway (2018) notes, obliging leaners to demonstrate progression by frequently taking level-based exams can have a major effect on motivation. For such learners, who are often desperate to improve their English language communication skills yet fearful of a system of rigorous testing and frequent assessment, an alternative approach to language support is necessary.

Secondly, a more flexible approach would be welcomed by those who, having been away from formal education for a long time, now feel a sense of constraint at having to study, often with classmates much younger than themselves, and follow a curriculum that can feel far removed from the reality of their daily lives. Simpson (2016, p. 180). draws attention to the contexts, locations and experiences that constitute refugees' daily lives, reflecting that:

> " ... *institutions such as government employment offices, welfare offices and banks loom large in the lives of linguistic minority people and students' interactions in English can be coloured by miscommunication, hostility and sometimes racism*".

His observation supports the concerns voiced by other stakeholders who feel that ESOL provision should more closely reflect learners' immediate needs. For example, one resettlement officer felt that the biggest problems facing refugees on VPRS was learning how to deal with the UK Government's new, online welfare benefit system—the Universal Credit system. While these issues may be touched upon on a general English ESOL course, the demands of an externally set assessment schedule mean

that the topics cannot be addressed in the depth required to meet the students' contextual needs. We argue, therefore, that the type of ESOL education that is offered to refugees on VPRS schemes can be enhanced in all areas through designing provision that more closely reflects their daily-life needs. Simpson (2016, p. 2) notes that on assessment-based accredited courses, "*Much ESOL content and teaching material ironically does not actually prepare students for the real-world challenges they face*". This appears to hold true from the accounts recorded in our research. That is to say, it is not only the quantity of classroom opportunities that is important to ESOL learners, but also the content and methodology of the classes.

*5.3. Alternative Options*

For the reasons referred to above, we suggest that traditional models of ESOL progression, which can take years to complete, fall short of meeting the linguistic needs for VPRS participants, many of whom arrive in the UK with pre-entry level English (A1 CEFR). The current guidance recommends a model of ESOL delivery that involves placement on formal, accredited courses for "quality assurance and value for money" (Home Office et al. 2016, p. 4). Such classes, necessarily, follow a pre-determined, prescribed syllabus with regular tests and assessments. What is more, the findings discussed in this paper highlight how meeting the government guidelines for recommended ESOL provision is extraordinarily difficult given the complexity of the learner cohort.

We argue that the case study data drawn upon here strongly suggest that a greater emphasis should be placed on funding and supporting approaches to language learning that encourage a sharper focus on learners' actual lives, their daily encounters and immediate linguistic needs. What is required, is a form of language education that does not solely oblige students to experience a centralised curriculum and assessment regime.

One model of delivery that supports a less syllabus-driven approach to ESOL classes is put forward by advocates of participatory pedagogy. Inspired by the work of Paulo Freire (1972) it is a problem-posing, dialogic approach that encourages students to:

> "*Take action on the issues which they identify as important, and evaluate their progress and the effectiveness of their programme as they go along*". (Cooke et al. 2015, p. 215)

Whereas conventional ESOL classes are based on a primarily linguistic syllabus guided by the schema of a course-book or externally set exam criteria, participatory ESOL focuses on issues that the students themselves have identified as important. Research into refugee integration programmes in Finland (Intke-Hernandez 2015) describes how conventional language programmes do not suit all migrants. Intke-Hernandez reports on a case study in which Finnish language teachers abandoned many of their prescribed teaching materials and methods and adopted a Freirean-type pedagogy. The approach allowed mothers to "*dictate the pace of the meetings and even the subject areas that were discussed . . . they took over the space*" (ibid., p. 126). The research draws attention to the unpredictability of adult language-learning needs and how, regarding integration, there are strong arguments for intertwining classes with the needs of the learners and life in the surrounding residential area and society.

In a participatory pedagogy, the content and linguistic focus of the classes therefore emerges from class to class depending on the topics, themes and areas co-negotiated between the teacher and learner. Proponents of participatory ESOL have put forward a selection of techniques, tools and activities to promote such an emergent syllabus arguing that the approach allows classes to:

> "*Provide a challenging but safe environment for critical discussion which start by exploring the students' own ideas, thoughts, and experiences, gradually moving into discussion of ideas drawn from outside*". (Cooke et al. 2015)

Such a methodology appears a good fit for people on VPRS (and for many refugees in general), for whom years of accredited, college-based instruction, as we have contended, would unlikely be effective. Simpson (2019, p. 12) argues that "*responsive adult migrant language education will reflect the*

*domains of practice where migrants are actually present.*" Domains may include matters such as applying for travel documents, supporting family visa applications, applying for student funding, or as mentioned here, interacting with the Universal Credit system. Providing support for participatory ESOL classes, included as part of a suite of English language provision, would, logically, go some way toward mitigating some of the dissatisfaction expressed by the participants on the VPRS. For example, having greater agency in the classroom might well assuage some of the anxieties that participants reported about feeling as if they were in school once again. Classes would have more freedom to focus on issues that matter the most to the participants—such as discussing employment-based topics, communicating with schools or welfare system processes. For mothers with children, as seen in the example from Finland, such an approach might well be possible to enact in a family learning environment, with no pressure of preparing for exams or tests.

### 5.4. Concluding Remarks

The Government's policy for Wales acknowledges the complexity involved in organising language education and recommends that funders attempt to:

> "Offer provision to meet the variety of needs of learners, taking into account: appropriate levels, location, timetables and the educational backgrounds of learners". (p. 16)

It has been argued here that a more flexible suite of ESOL provision would improve both the experience of language learning and integration for VPRS participants and may also be suitable for many other people seeking sanctuary in Wales. However, in order to facilitate an enhanced ESOL provision, change and development are required. For example, models of pre-service and in-service teacher education that include specific focus on ESOL teaching and learning, rather than on general English language teaching, are now needed. In addition, available funding for language learning should acknowledge and support language classes delivered using non-traditional methodologies and pedagogies, where appropriate and necessary. As regards governance, further clarity is required as to who holds responsibility for ensuring that ESOL delivered to resettled refugees across Wales is consistent in terms of quality and quantity. With regard to further research, investigation into the viability and efficacy of alternative models of provision, such as those discussed above, would provide necessary insights into alternatives to current models. Moreover, research mapping the linguistic progression of people on VPRS would be valuable in informing future policy decisions around the funding and organisation of ESOL provision.

**Author Contributions:** Conceptualization, M.C. and I.H.-L.; methodology, M.C. and I.H.-L.; formal analysis, M.C. and I.H.-L.; investigation, M.C. and I.H.-L.; data curation, I.H.-L. and M.C. writing—original draft preparation, M.C.; writing—review and editing I.H.-L. and M.C.; supervision, M.C.; project administration, M.C. and I.H.-L.; funding acquisition, M.C. and I.H.-L.

**Funding:** This research was funded by Knowledge Economy Skills Scholarships (KESS). KESS is a pan-Wales higher-level skills initiative led by Bangor University on behalf of the HE sector in Wales. It is part funded by the Welsh Government's European Social Fund (ESF) programme for East Wales. Funding was also provided by the Welsh Refugee Council. [C80815 KESS 2]

**Acknowledgments:** The authors would like to thank all the participants who gave their time and energy in helping us gather the data for this project. We would also like to thank the Welsh Refugee Council and the University of South Wales for supporting us wholeheartedly in carrying out this investigation.

**Conflicts of Interest:** The authors declare no conflicts of interest.

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
