# Peer review of "Language Education for Forced Migrants: Governance and Approach"

_languages, doi:10.3390/languages4030074_

Round 1

Reviewer 1 Report

Very interesting discussion about complexities involved in designing, organising and delivering language education programs for refugees on the Syrian Vulnerable Persons’ Resettlement Scheme (VPRS).

The data is relevant, and might be a start for further research, not limited to Wales and in the long run. 

The discussion falls a bit short but interesting, claiming language programs with a more flexible and participatory pedagogy seem important and updated pedagogy. 

Author Response

We thank the reviewer for taking the time to examine our paper. We have now looked again at all spelling issues and also noted a missing reference which jas been inserted.

Reviewer 2 Report

The author/s have done a very good job in their research, especially considering the difficulty in access to interviewees under the conditions of migrant/refugee situation, especially when it concerns female subjects. As a minor comment, I would add, however, that when reading about the situation of the studied female subjects, I was wondering whether women get support in their desire for English skills from male members of their families, or, rather, this is yet another difficulty to overcome.

The conclusions, calling for more flexible approach in provision of English as a foreign language to migrants and a more participatory pedagogy, are highly sound and should reach policy makers and educators. The article shows the importance of the socio-cultural context of teaching English as a foreign language. This context can be more fundamental than any classroom procedure and, when ignored, defeat the best curriculum and assessment regime.

Author Response

We would like to thank the reviewer for taking the time to examine our paper - and we are very happy that the reviewer found our research interesting and our conclusions relevant. 

We have now looked at all spelling issues. Regarding women's support for access to language classes, from their own families, although we did not collect any quantitative data especially on this issue, we found that the men generally did support their partners in attending classes. However, in situations where a family had young children, in most cases we encountered it was most unlikely to be the man that remained at home while the woman attended the class. 

Thank you again for taking the time to examine our paper - we would be happy to answer any further questions.